# Cell Anomaly Localisation using Structured Uncertainty Prediction Networks

**Boyko Vodenicharski** *University of Bath*      BIV20@BATH.AC.UK

**Samuel McDermott** *University of Cambridge*      SJM263@CAM.AC.UK

**Katherine Webber** *University of Cambridge*      KMW59@CAM.AC.UK

**Viola Introini** *University of Cambridge*      VI211@CAM.AC.UK

**Pietro Cicuta** *University of Cambridge*      PC245@CAM.AC.UK

**Richard Bowman** *University of Bath*      RWB34@BATH.AC.UK

**Ivor J. A. Simpson**[*] *University of Sussex*      I.SIMPSON@SUSSEX.AC.UK

**Neill D. F. Campbell**[*] *University of Bath*      N.CAMPBELL@BATH.AC.UK

**Editors:** Under Review for MIDL 2022

## Abstract

This paper proposes an unsupervised approach to anomaly detection in bright-field or fluorescence cell microscopy, where our goal is to localise malaria parasites. This is achieved by building a generative model (a variational autoencoder) that describes healthy cell images, where we additionally model the structure of the predicted image uncertainty, rather than assuming pixelwise independence in the likelihood function. This provides a "whitened" residual representation, where the anticipated structured mistakes by the generative model are reduced, but distinctive structures that did not occur in the training distribution, e.g. parasites are highlighted. We employ the recently published Structured Uncertainty Prediction Networks approach to enable tractable learning of the uncertainty structure. Here, the residual covariance matrix is efficiently approximated using a sparse Cholesky parameterisation. We demonstrate that our proposed approach is more effective for detecting real and synthetic structured image perturbations compared to diagonal Gaussian likelihoods.

**Keywords:** Generative models, variational autoencoders, out-of-distribution detection, structured uncertainty

## 1. Introduction

Curating datasets of microscopy images for scientific or clinical study is a time consuming task that often requires expert users; throughout this work we consider the localisation of malaria parasites of the species *plasmodium falciparum* in thin film blood smears (Rajaraman et al., 2018). The majority of the cells in a smear of infected blood will appear healthy with the exception of roughly between 0.1 and 4% (this number is called *parasitemia*) of cells which are infected, which can be in different stages of infection, all of which have an unique "anomalous" look. The parasites tend to synchronise their lifecycle stages and the parasitemia cycles between low and high numbers during the course of the illness. When the parasitemia is low, malaria is very hard to spot, which complicates the labelling and diagnostic challenge since a human annotator must search exhaustively through many images (fields of view), carefully examining each cell to identify if it is infected.

---

[*] Contributed equally

Our goal is to build an unsupervised learning system that can identify and localise potential anomalies in cell microscopy data. We envisage this will facilitate efficient selection of data to be labelled for tasks such as segmentation or classification, and is a step towards *active learning* for such problems (Settles, 2012). Importantly, healthy blood cells are both easy to image and available in a large volume, while infected images are harder to acquire.

Our approach is posed as learning a probabilistic generative model of healthy cell appearance. Since we learn a distribution over images, we can establish whether previously unseen data is well explained by the healthy model, or if it has an unexpected appearance. The novelty in this work is we employ the structured uncertainty prediction networks (SUPN) approach to specify a structured distribution over the *residual image*, the difference between the model prediction and observation. This structure enables us to create a "whitened" residual representation that removes expected correlations, providing an interpretable quantification of the departure from the expected appearance.

In this paper we examine both bright-field and fluorescent microscopy data. We employ existing cell segmentation tools (Stringer et al., 2020) to provide cropped images of cells in uncluttered regions to our model. We do not perform any manual data curation.

In summary, the contributions of the paper are: (i) a description of a new approach for localising anomalous image components based on structured uncertainty; (ii) a validation of the benefits of structured uncertainty when identifying synthetic artefacts in bright-field imaging; and (iii) a demonstration of detecting malarial parasites on fluorescent microscopy.

**Related Work**  Work on unsupervised anomaly detection largely belongs to two approaches. The first involves modelling the "healthy" data distribution using a generative model, then checking how large of a discrepancy there is upon reconstructing the query datapoint, using a specific metric. Our model is of this type, and for a review, see (Baur et al., 2020). The other approach involves using a pretrained network, usually on ImageNet, as a feature extractor and then estimating a distribution of the data in the feature space, and then looking for outliers. The leading models on the MVTec industrial dataset belong to this category, see (Yu1 et al., 2021), (Cohen and Hoshen, 2021), (Roth et al., 2021), (Defard et al., 2020). To the best of our knowledge, we are the first to apply an unsupervised approach to red blood cells and malaria, but well-performing supervised methods exist (Rajaraman et al., 2019).

## 2. Background

**Learning image distributions**  Deep probabilistic generative models, such as the variational autoencoder (VAE) (Kingma and Welling, 2014; Rezende et al., 2014), realNVP (Dinh et al., 2016) and GLOW (Kingma and Dhariwal, 2018) provide frameworks for learning distributions of images. Given such a trained model, one would anticipate being able to estimate the probability that a new image is drawn from the training distribution. However, it has been demonstrated that, if applied naively, such models are surprisingly ineffective at out-of-distribution detection (Nalisnick et al., 2019) tasks as they learn local image statistics without linking them to global image properties. Subsequent work partially addressed this issue by considering likelihood ratios that compensate for the complexity of observed data and the training distribution (Ren et al., 2019; Serrà et al., 2019).

Importantly, most prior work has considered large scale distribution shift (e.g. MNIST to SVHN); whereas we are investigating more subtle anomalies, where the input image complexity is unlikely to vary substantially between the in/out distributions. Accordingly, we expect (and demonstrate) that generative models can be useful tools in our scenario.

In this paper we limit our investigation to models that assess data probability in the observation domain, such as the VAE. Given such a trained model, one approach to detect out-of-distribution examples is to measure the distance between the closest point(s) on the learned distribution and the candidate image to assess similarity. However, this requires a suitable metric between images; typically this metric accords with the likelihood function used to train the model, but other metrics are available as discussed in appendix D.

**Variational AutoEncoders**  The VAE (Kingma and Welling, 2014; Rezende et al., 2014) is a probabilistic generative model; it predicts an approximate distribution for latent variables, $q_\phi(\boldsymbol{z}|\boldsymbol{x})$, given input data, $\boldsymbol{x}$, where $q_\phi(\cdot)$ is a distribution with parameters determined by a neural network with weights $\phi$. $q_\phi(\cdot)$ generally takes the form of a multivariate Normal (MVN) distribution with a diagonal covariance matrix, i.e. $q_\phi(\boldsymbol{z}|\boldsymbol{x}) = \mathcal{N}\big(\boldsymbol{\mu}(\boldsymbol{x}), \boldsymbol{\sigma}^2(\boldsymbol{x}) \circ \boldsymbol{I}\big)$. The encoder is jointly trained with a decoder $p_\theta(\boldsymbol{x}|\boldsymbol{z})$, which maps from a latent vector $\boldsymbol{z} \sim q_\phi(\boldsymbol{z}|\boldsymbol{x})$ to predict a distribution over the observation domain. Again, the parameters of this generating distribution $p_\theta(\cdot)$ are determined by a neural network with weights $\theta$. In most incarnations, $p_\theta(\boldsymbol{x}|\boldsymbol{z})$ also takes the form of a MVN:

$$p_\theta(\boldsymbol{x}|\boldsymbol{z}) = \mathcal{N}\big(\boldsymbol{x} \mid \boldsymbol{\mu}(\boldsymbol{z};\theta), \boldsymbol{\Sigma}(\boldsymbol{z};\theta)\big). \tag{1}$$

Commonly $\boldsymbol{\Sigma}$ is a diagonal matrix $\boldsymbol{I}$ multiplied by a learned or fixed scalar, often termed the "$L2$-loss", although it can also be heteroscedastic (per-pixel variances).

A unit Normal prior is used to encourage regularity in the latent space, $p(\boldsymbol{z}) = \mathcal{N}(\boldsymbol{0}, \boldsymbol{I})$. Following the standard variational inference formulation (see e.g. (Blei et al., 2017) for a derivation) we arrive at the evidence lower bound (ELBO, $\mathcal{L}$):

$$\log p(\boldsymbol{x}) \geq \mathcal{L} = \underbrace{\mathbb{E}_{q_\phi(\boldsymbol{z}|\boldsymbol{x})}\big[\log p_\theta(\boldsymbol{x}|\boldsymbol{z})\big]}_{\text{Likelihood Term}} - D_{\mathrm{KL}}\big[q_\phi(\boldsymbol{z}|\boldsymbol{x}) \,\|\, p(\boldsymbol{z})\big], \tag{2}$$

Training involves maximising $\mathcal{L}$ with respect to the encoder and decoder weights $\phi$ and $\theta$. For simplicity, we illustrate the ELBO with a single example, $\boldsymbol{x}$, whereas in reality this objective should be evaluated over the training dataset. Practically, $-\mathcal{L}$ is averaged over examples in mini-batches of the dataset following a stochastic training procedure.

Unsupervised anomaly detection using autoencoders has been investigated on brain MRI, with a comprehensive overview given in (Baur et al., 2020). Here, they established that a restorative VAE, which additionally uses test-time optimisation of the latent representation, provides the best performance. Such advances are complementary to our presented research, and we demonstrate they can be used in combination with our approach.

**Structured Likelihoods**  The likelihood term, identified in Equation (2), requires the model to explain the training dataset, and provides a distance between the predicted mean and the observations. Commonly the residual error are assessed independently for each pixel. However, it is well known that image data lie on a low dimensional manifold due to strong pixelwise correlations (Mallat, 1999). Accordingly, the assumption of spatial

independence in the residual is demonstrably false; significant structure is preserved relating to spatially connected components that are incorrectly predicted

This work uses a likelihood model that explicitly captures pixelwise correlations. Modelling structured Gaussians with neural networks was investigated historically for low-dimensional data (Williams, 1996). However, fitting a dense covariance MVN to image data is a difficult task, even for small images, due to the quadratic growth of $\boldsymbol{\Sigma}$ with the number of pixels. A tractable solution for image data was recently proposed termed Structured Uncertainty Prediction Networks (SUPN) (Dorta et al., 2018). This approach approximates the covariance matrix using a sparse Cholesky decomposition of its' inverse, i.e. $\boldsymbol{\Sigma} = \boldsymbol{\Lambda}^{-1} \approx (\boldsymbol{L}\boldsymbol{L}^{\top})^{-1}$. The sparsity pattern links pixels within a local neighbourhood. We can rewrite the log-likelihood of Equation (1) as:

$$\log \mathcal{N}\big(\boldsymbol{x} \,\big|\, \boldsymbol{\mu}(\boldsymbol{z}), \boldsymbol{\Sigma}(\boldsymbol{z})\big) = \frac{1}{2}\log\big(2\pi|\boldsymbol{L}\boldsymbol{L}^{\top}|\big) - \frac{1}{2}(\boldsymbol{x}-\boldsymbol{\mu})^{\top}(\boldsymbol{L}\boldsymbol{L}^{\top})(\boldsymbol{x}-\boldsymbol{\mu}), \tag{3}$$

where we have dropped the dependence on $\theta$ for simplicity; these neural network weights will determine the predictions for $\boldsymbol{\mu}(\boldsymbol{z};\theta)$ and $\boldsymbol{L}(\boldsymbol{z};\theta)$.

We note that despite $\boldsymbol{L}$ being sparse, the approximated covariance $(\boldsymbol{L}\boldsymbol{L}^{\top})^{-1}$ is dense; long-range correlations are possible even with small neighbourhood connections in $\boldsymbol{L}$ (e.g. Figure 6), as long as there are connected pixels that are correlated. A more detailed explanation of the approximation can be found in Dorta (2020).

Finally, we note that the Cholesky parameterisation has another appealing property: we can use it to "whiten", decorrelate and descale, the residual difference image, $\boldsymbol{L}(\boldsymbol{x}-\boldsymbol{\mu})$. This leads to a normalised residual image in which the Euclidean distance is more appropriate, see Figure 1 for an example. The predicted covariance structure explains the residual correlations across the image domain that are expected given the training dataset, and reduces the "cost" of anticipated mistakes. However, departures from the expected structure in the residual lead to a large distance. As such, this whitened residual space provides a useful domain in which we can localise anomalous features in the image.

**Relationship to the Mahalanobis distance**   The Mahalanobis distance (Bishop, 2006) is a distance metric for a vector under a MVN distribution:

$$D_{\mathrm{M}}(\boldsymbol{x}) = \sqrt{(\boldsymbol{x}-\boldsymbol{\mu})^{\top}\boldsymbol{\Sigma}^{-1}(\boldsymbol{x}-\boldsymbol{\mu})}. \tag{4}$$

We can consider this as an alternative metric to the MVN likelihood of Equation (1).

## 3. Method

**Model**   We use a VAE architecture that consists of a single encoder and separate decoders for predicting the mean and the covariance estimates, either a single channel for the uncorrelated Gaussian, or multiple channels for the Cholesky matrix $\boldsymbol{L}$; this is illustrated in Figure 3. By definition, the diagonal entries in $\boldsymbol{L}$ are positive, so we predict their log-values.

**Training**   Training is separated into two stages. First we assume a spherical covariance matrix (constant global variance) and train only the Encoder and $\boldsymbol{\mu}$-Decoder, while the $\boldsymbol{L}$-Decoder is ignored. The empirical variances were $\sigma_{\mathrm{BF}}^2 = 0.06$ for brightfield (Dataset

1) and $\sigma_{\mathrm{FL}}^2 = 0.11$ for the fluorescent (Dataset 2) datasets. The first stage loss, ignoring constant terms, is written as:

$$\mathcal{L}_1 = \frac{1}{\sigma^2}(\boldsymbol{x} - \boldsymbol{\mu})^2 + D_{\mathrm{KL}}\big(q(\boldsymbol{z}) \,\|\, p(\boldsymbol{z})\big). \tag{5}$$

In the second stage, we fix the Encoder and $\boldsymbol{\mu}$-Decoder, and train the $\boldsymbol{L}$-Decoder using the negative log-likelihood of Equation (3). The second stage loss is written as:

$$\mathcal{L}_2 = -\log\mathcal{N}\big(\boldsymbol{x} \,\big|\, \boldsymbol{\mu}, (\boldsymbol{L}\boldsymbol{L}^\top)^{-1}\big) + D_{\mathrm{KL}}\big(q(\boldsymbol{z}) \,\|\, p(\boldsymbol{z})\big), \tag{6}$$

Both stages are run for approximately 100,000 iterations with a batch size of 256. In the first stage, the learning rate starts off at 0.001 and is reduced by a factor of 10 halfway through. In the second stage, the learning rate is kept at 0.001 throughout the training. In all cases we use the Adam optimizer (Kingma and Ba, 2017). The two-stage training process is required to prevent the model "cheating" and describing all the image information through $\boldsymbol{L}$, which has more parameters than $\boldsymbol{\mu}$.

As with all deep models, we note a sensitivity to initialization. We perform 1-4 random initializations until a good solution was found. Work has been done to develop robust VAE optimisation methods (Rezende and Viola, 2018), which we may explore in future work.

**Learned Mahalanobis Distance**   The VAE is trained to represent the healthy reference images in its latent space. Accordingly, the predicted distributions will have limited understanding of out-of-distribution examples. Test images are projected to the encoding manifold in the latent space; any anomalous image will be represented only using reference "healthy" features. An encoded image is captured by a distribution in the latent space, so we calculate the expected value of the Mahalanobis distance

$$D_M(\boldsymbol{x}) = \mathbb{E}_{\boldsymbol{z} \sim q_{\boldsymbol{z}}}\big[D_M(\boldsymbol{x}, \boldsymbol{z})\big] \approx \frac{1}{S}\sum_{s=1}^{S} D_M(\boldsymbol{x}, \hat{\boldsymbol{z}}_s), \tag{7}$$

by averaging over $S$ samples $\hat{\boldsymbol{z}}_s$ drawn from the latent distribution $q(\boldsymbol{z} \,|\, \boldsymbol{x})$.

## 4. Experiments and Results

In all of our experiments, we apply a Gaussian blur of kernel size 5 and sigma 2 to the images in all models in addition to keeping the scale within $(-1, 1)$ of the inputs.

**Data**   Our experiments use two datasets, both with human red blood cells (RBCs) roughly centered in a 128 pixel-wide square field of view. Dataset 1 contains brightfield images of a sample of healthy live RBCs, in a dilution that allows the cells enough space to not overlap significantly. Dataset 2 consists of SYBR Green I tagged malaria parasites of the species *plasmodium falciparum* in a fixed sample. The RBCs are autofluorescent so appear as bright regions in the images. This dataset is much noisier due to the higher gain of the camera, as the signal is much weaker than in brightfield imaging. Both datasets were imaged using the same inverted microscope, in brightfield and fluorescence imaging modes respectively.

We used Cellpose (Stringer et al., 2020), a generalist cell instance segmentation network, to produce cropped fields of view around the cells. Dataset 2 had to be hand-curated to

separate the infected cells for evaluation; this was performed by a user with prior experience of detecting parasites under fluorescence imaging. Low parasitemia resulted in a smaller dataset size. Overall, Dataset 1 consists of 90,345 individual cell images, and Dataset 2 consists of 1,316. Results are calculated on a holdout set of 500 and 100 healthy cells respectively; dataset 2 also includes 100 infected cells. All images are rescaled to $(-1, 1)$.

**Samples**   Figure 6 shows the fit of the trained SUPN model on test brightfield images from dataset 1. Samples were taken and ranked, by hand, to represent increasingly challenging images in terms of the complexity of the cell including neighbouring cells and overlapping cells. We observe that the (log) variance recovered by the model shows that the residual structure is expected by the model; high residual regions are supported by regions of high variance. We also draw some random samples from the recovered MVN distribution to demonstrate the structure that has been captured by the model. In particular we can see the ambiguities over cell boundaries when multiple cells border and overlap. Importantly, this allows us to observe what the model has learned; we note that drawing such distributions from spherical or diagonal models would simply add independent noise per pixel and demonstrates that structure in the residuals has not been learned.

**Localisation via Whitened Residuals**   By calculating the pixelwise contribution to the Mahalanobis distance, we can create a *whitened residual* images. Areas that are well-explained have low values and unexplained features are bright. The top of Figure 1 demonstrates this effect; where outlier features are visible in the the whitened residual. Our experiments illustrate two scenarios: adding artificial outliers to healthy brightfield images, or examining fluorescent images that contain real parasites.

Figure 2 provides localization examples for real parasites in the fluorescent dataset 2. This dataset is challenging due to the high noise levels in the raw images, which can be seen in the difference between the raw and predicted mean images. The final column shows random structured noise samples, which visually appear similar to the input data. This illustrates that the noise distribution has been well captured. The high level of noise means that the spherical model is unable to separate the parasites from the noise floor. Equally, the diagonal model is overly sensitive since the independent, per-pixel noise estimates are not well matched. Our SUPN model is better able to trade-off the unstructured sensor noise and the structured anomalies in the image making it easier to detect the parasites even in these challenging examples. We quantify the detection performance on this data in Table 1. Establishing a strong baseline was difficult, to the best of our knowledge we are the first to apply unsupervised anomaly detection on red blood cells. We include comparisons with high-performing approaches on on the MVTec dataset(Paul Bergmann, 2021). However, these methods perform poorly due to the domain shift from industrial to microscopy data.

**Perturbing Images with Structured Noise**   Calculating the Mahalanobis distance for the scene is a very simple indicator of whether an anomaly is present. In this experiment, we introduce an ellipsoidal anomaly inside a cell, in a controlled way, and check how the (expected) Mahalanobis distance changes for each distributional assumption, as we vary the alpha value (transparency) of the anomaly. The ellipsoid is located within the cell and either filled with Gaussian white noise (Noisy Ellipsoid) or a smoothly varying distortion (Smooth Ellipsoid); Figure 4 in the appendix illustrates the effect of the $\alpha \in [0, 1]$ parameter.

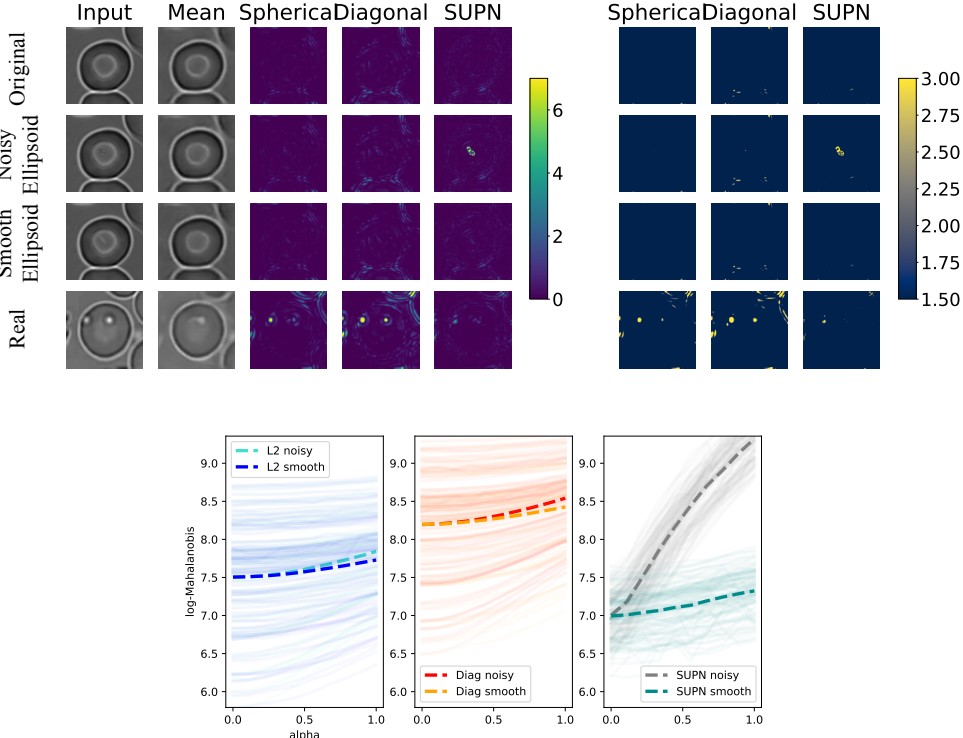

Figure 1: Illustration of the whitened residuals for each model for localisation. Top: We show results for the three models on (top to bottom) an unadulterated test image, two synthetic test cases (an artificial ellipsoid added via iid Gaussian noise or a smooth perturbation), and a real example (all locations highlighted by green arrows). The whitened residuals for the three models are provided with a full scale and a reduced scale to illustrate localisation. Bottom: graphs of metric values vs noise amplitude. The transparent lines in the background represent the response for an individual image, for 100 different images (the anomalies are all placed in the same spot). The broken line is the mean. A visualization of the effect of the alpha parameter (only shown for alpha=1 here) can be seen in the appendix. In the first graph L2 refers to the Spherical Gaussian distribution and the two mean lines are superimposed.

The result of this experiment is shown in the bottom of Figure 1. Rather than comparing the absolute values between methods, we consider the difference between the two artificial ellipsoids; the noisy vs the smooth. Individual examples from the test data set are provided as transparent lines (to estimate the variation in the results) with the bold, broken lines showing the mean across 100 test images.

We see from the comparison between the different types of artefacts that the Spherical model makes no distinction between types of noise. The diagonal model reacts slightly, but the SUPN model clearly separates the two types of anomaly. Importantly, we can see that SUPN approach has learned the structure that is to be expected within the cell and independent noise in the centre of the cell should be unlikely but the per-pixel approaches of the other approaches cannot separate this from the noise floor. In fact, the spherical model is never able to separate the two whereas the diagonal model succeeds at around $\alpha = 0.5$.

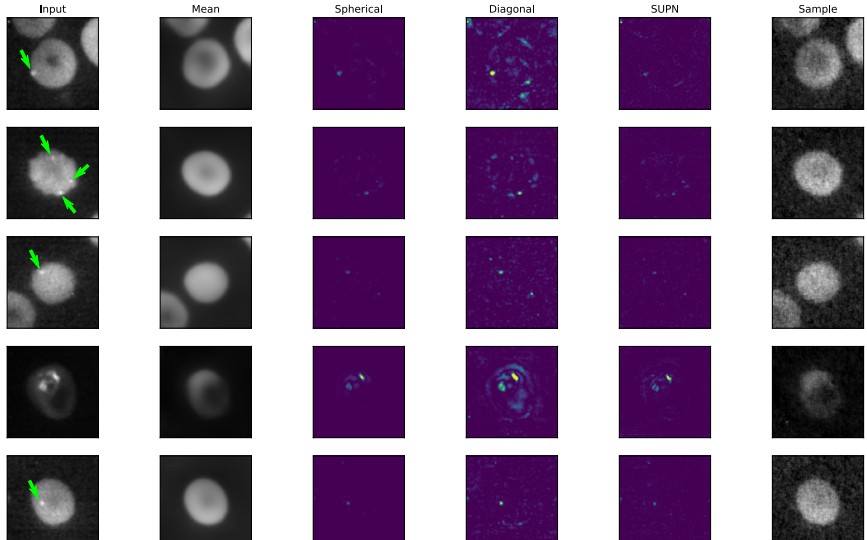

Figure 2: We show examples of the model trained on fluorescent microscopy images. The middle three rows show the whitened residuals, where bright parts can be interpreted as outlier features. The parasites can be seen as bright spots in the raw input images (highlighted by green arrows), most clearly visible as the bright pattern. We also include a random sample from the SUPN model illustrating that the noise statistics are well modelled.

## 5. Conclusion

Previous works have illustrated that deep generative models are generally ineffective at out-of-distribution detection (Nalisnick et al., 2019), and we find this is indeed true for diagonal covariance matrices with even the perturbations we employ here. We demonstrate that the use of structured residuals enables the model to detect certain out-of-distribution examples. However, in this application the test and train distributions had really minor differences, and we have not evidenced any capability to detect large distributional shifts.

Whilst our experiments are limited to optical microscopy, the framework presented is general and more widely applicable. Local correlations are known to exist across a range of imaging modalities and many existing approaches do not make use of likelihood models that account for this residual structure in the observations. We hope that this work opens further investigation and will yield wider adoption and development of works using structured uncertainty.

## Acknowledgments

We would like to acknowledge funding from EPSRC through grants (EP/R011443/1), (EP/R013969/1) and the CAMERA research centre (EP/T022523/1), as well as from the Royal Society.

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

## Appendix A. Model architecture

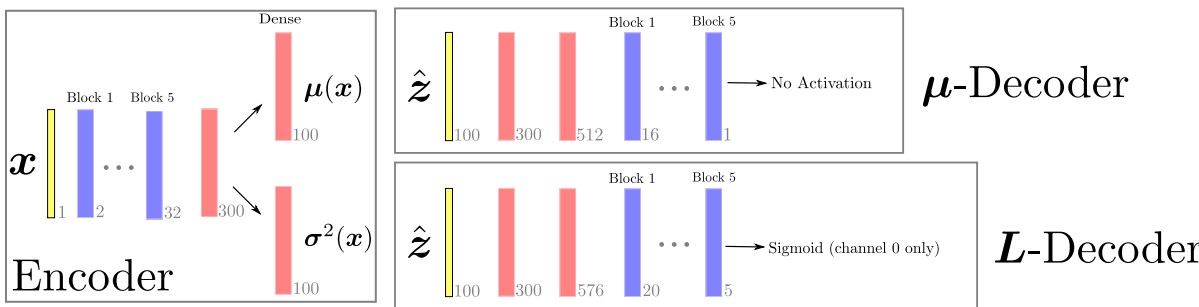

Figure 3: An illustration of our VAE encoder-decoder architecture. Input, $\boldsymbol{x}$ is a single channel square image of 128 pixels width. The convolutional blocks in the Encoder consist of two Conv2D operations, followed by a Pool2D and a ReLU activation. The channels double with each block, while the image size is halved. The final 3 blocks in the Encoder are fully connected and produce the mean $\boldsymbol{z}$ and log-diagonal variance vector $\boldsymbol{\Omega}$. A latent sample $\hat{\boldsymbol{z}} \sim q(z)$ is input to the decoders. The convolutional blocks of both decoders use upsampling operations instead of pooling. The log diagonal of $\boldsymbol{L}$ ($0^{\text{th}}$ channel of the $\boldsymbol{L}$-Decoder) is transformed by a scaled and shifted sigmoid activation to limit the predicted precision. The other channels, i.e. the offdiagonal entries in $\boldsymbol{L}$ and predicted mean $\boldsymbol{\mu}$-Decoder have no output activation All kernel sizes are $3 \times 3$.

## Appendix B. Effect of alpha value

Figure 4 illustrates the smooth blending of the ellipsoids in the synthetic anomaly experiment.

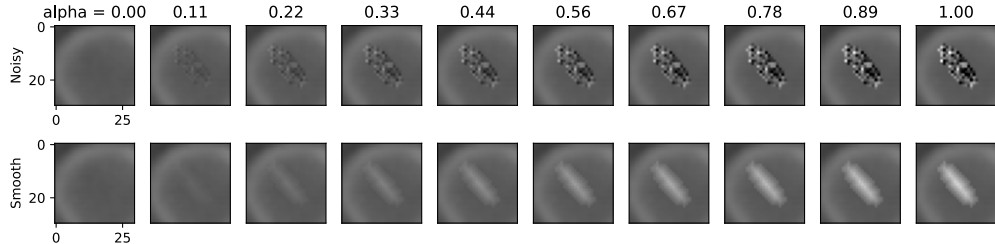

Figure 4: Effect of the alpha blend value when introducing localized structured and unstructured noise.

## Appendix C. Challenging Example for Whitened Residuals

Figure 5 provides a really challenging scenario for localisation due to a large number of bordering cells and significant overlap in both the synthetic case and the real example.

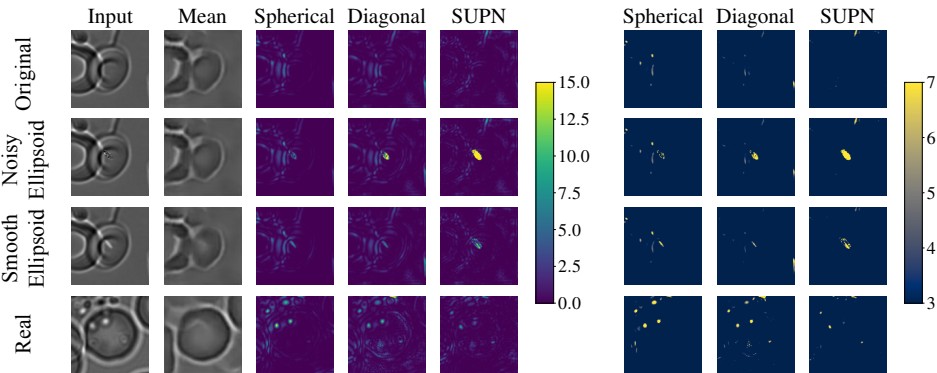

Figure 5: Additional challenging illustration of the whitened residuals for each model for localisation. We show results for the three models on (top to bottom) an unadulterated test image, two synthetic test cases (an artificial ellipsoid added via iid Gaussian noise or a smooth perturbation), and a real example. The whitened residuals for the three models are provided with a full scale and a reduced scale to illustrate localisation. Here overlapping cells in both the synthetic and real examples make detection and localisation more difficult.

## Appendix D. Relation to Learning Metrics between Images

The challenge presented here is quite specific in terms of providing a ranking metric that is tailored to the particular microscopy images. At one end of the scale we have deterministic metrics (e.g. the L2 norm between images) that are easy to evaluate but known to present a poor approximation to perceptual differences as they treat each pixel independently; they fail to model any structure in the image.

At the other end of the spectrum, there are learning-based metrics that seek to account for general perceptual distances; these may be separated into two groups. The first set of approaches, such as the Fréchet inception distance (Heusel et al., 2017), form a metric using a representation obtained from some associated computer vision task. Such models look collectively at a global pixel structure but are based on features extracted from networks trained on high level content classification, e.g. object recognition through datasets such as imagenet. To this end, these measures are not selective for our specific task where the differences are subtle are require expertise and experience for the human annotator. Importantly, we do not have access to large labelled datasets of microscopy images as the creation of such datasets is the motivation for this work.

**Deep Metric Learning** The second set of approaches attempts to learn a metric through explicit training. For example, deep metric learning approaches, (Suárez et al., 2021), require curated training data. Since it would be difficult to provide an explicitly labelled distance between training images a-priori (i.e. to treat metric learning as a regression task), data is usually annotated in the form of triplets as a reference image with a positive (nearby) and negative (far away) example. Learning approaches seek to ensure that similar images are spaced closely whilst dissimilar ones are guaranteed to separated by more than some minimum threshold. This is a challenging learning problem as the negative examples are unbounded, there is a weak learning signal for detailed metrics (e.g. the difficulties between

inter- and intra-class variation), and the triplet datasets still require annotation. Important recent work from Musgrave et al. (2020), has demonstrated that great care is required in the formulation of such metric learning; there are still a lot of unanswered questions around the efficacy and in terms of which approaches to use.

In contrast to these approaches, we propose an unsupervised approach to learning a metric that can be trained specifically for the normal case that allows for out-of-distribution detection of the rare anomalies. We train a generative model to recreate the common case dataset by capturing its statistics; this ensures our metric is tailored to the particular images without requiring any labelling of anomalies.

Put into the context of previous work:

- Unlike deterministic methods, e.g. L2, we capture local structure in the images, rather than treating pixels independently, and are therefore able to differentiate between noise and structured anomalies.

- Unlike the first set of learning metrics, e.g. FID, we do not required annotations for an associated task on the same dataset to obtain a metric tailored to our particular images.

- Unlike the second set of explicit metric learning, we do not require annotation triplets; we are able to detect anomalies through an out-of-distribution process rather than an explicit set of labels.

## Appendix E. Samples

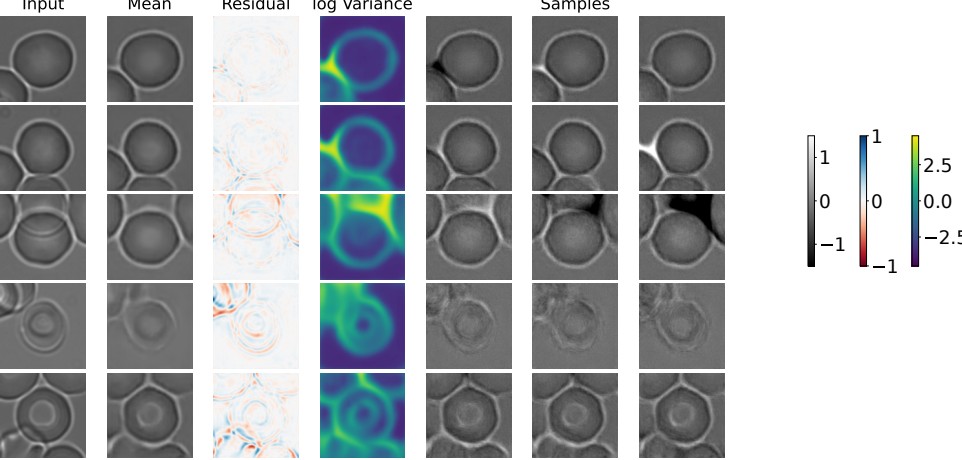

Figure 6: Illustrations of fit to test data and samples from the SUPN model. The (log) variance recovered by the SUPN model correlates well with the residual images and we show three random samples from the model to illustrate the range of variations that the model expects. Examples are increasingly challenging from top to bottom.

## Appendix F. Quantitative Results

To get the classification and localization results, we tile the cell (mask provided from Cellpose) into 4x4 regions. This size is chosen according to the size of the parasite features as we expect them to show up in the whitened residual. We sum the Mahalanobis distance within each region and use the maximum across regions to represent the final anomaly score for each image. We use these scores to calculate the area under ROC and PRC curves, which is shown in Figure 7. We attempted to use some of the leading models from the MVTec leaderboard directly on our fluorescent dataset, but they did not perform well. These are shown in Table 1, but we must stress that these models were never designed to work on microscopy data, so are not a strong baseline for comparison. We initially ran our experiments using L1 regularization on the off-diagonal terms in the Cholesky matrix. The presence of this regularization seems to have little effect on the classification performance of the model, so we have left it out. We have not, however, thoroughly explored its effect on the sample generation.

| Model Name | AUPRC | AUROC |
|---|---|---|
| SPADE | 0.613997 | 0.5704 |
| PaDiM | 0.513182 | 0.4992 |
| PatchCore | 0.527472 | 0.5106 |

Table 1: The AUROC and AUPRC scores on the holdout test set from dataset 2 of some leading MVTec models. We used the implementation of SPADE (Cohen and Hoshen, 2021), PaDiM (Defard et al., 2020) and PatchCore (Roth et al., 2021) from https://github.com/rvorias/ind_knn_ad as a reference for an industrial anomaly detection framework applied on cellular data - it likely does not perform as well due to using pretrained features in a natural setting.

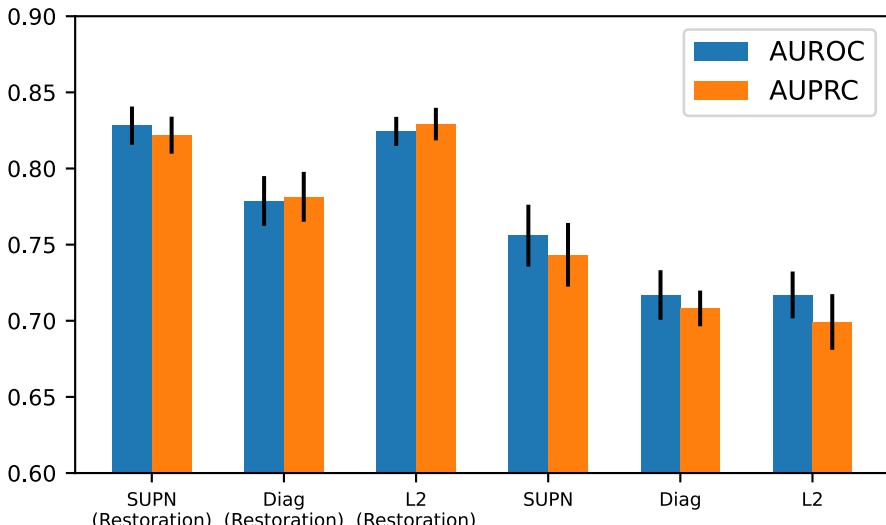

Figure 7: The AUROC and AUPRC scores on the holdout test set from dataset 2, using our VAE approach and different MVN models. The "Restoration" columns correspond to adding an additional gradient descent optimization step in the latent space when reconstructing the images. This is as suggested in (Baur et al., 2020) to provide the best performance in VAE models. The anomaly score on all VAE models was performed by tiling the image into 4x4 areas and summing the Mahalnobis distance within. For each image, the largest tile value was selected, and the results were given to the sklearn library to compute the area under the Receiver-Operating-Characteristics (AUROC) and the Precision-Recall-Curve (AUPRC). These metrics were selected because they are also used in (Baur et al., 2020) and higher is better. We also masked away regions outside of the cell, using the segmentation from CellPose, to make sure anomalies are restricted to inside the cell.

