# OpenReview forum: "Cell Anomaly Localisation using Structured Uncertainty Prediction Networks"
_MIDL.io/2022/Conference — MIDL 2022_

### Official Review · Reviewer_Jxwn · 2022-01-14

**Confidence:** 5
**Preliminary Rating:** 1
**Recommendation:** Poster

**Summary:**

The paper proposes to use a VAE for anomaly localization on cell microscopy images. The authors propose to parameterize p(x|z) with Structured Uncertainty Prediction Networks, explicitly with a Normal distribution parameterized by two "decoder" networks one for the mean and one for the variance (which here in contrast to most other works is predicted as a decomposed matrix instead of chosen as a constant).

**Strengths:**

- The paper is nicely written and structured.
- The application to cell microscopy images is to the best of my knowledge new.
- The use of Structured Uncertainty Prediction Networks for anomaly localization is a new and nice idea.

**Weaknesses:**

- The paper seems to ignore all previous anomaly localization and detection literature (see the MV-Tec Dataset, the MOOD MICCAI Challange as benchmark/datasets, "Autoencoders for unsupervised anomaly segmentation in brain MR images: a comparative study; C Baur, S Denner, B Wiestler, N Navab, S Albarqouni Medical Image Analysis, 101952" for a comparison paper in the medical area, "Unsupervised anomaly localization using variational auto-encoders; David Zimmerer, Fabian Isensee, Jens Petersen, Simon Kohl, Klaus Maier-Hein; International Conference on Medical Image Computing and Computer-Assisted Intervention"  for an overview of VAEs for localizing anomalies and the influence of the variance in p(x|z), or simple https://paperswithcode.com/task/unsupervised-anomaly-detection/latest for a lot of on-going work in the area).
- The paper gives no quantitative evaluation of the results and does not compare the results to any baseline.


**Deanonymize Review:**

no

**Detailed Comments:**

- In 3. Please explain how the empirical variances were chosen.
- The authors say, they perform 1-4 random initialization until a "good" solution was found. How is "good" defined and why sometimes one and sometimes 4?
- The dataset description can be improved. Does only dataset 2 include infected cells? What was dataset 1 then used for ? Only the noise models?
- I don't see why (if the absolute values can very), the margin in Figure 2 Bottom should give any qualitative superiority as long as the lines are separable (which for 2 lines, should almost always be the case). Overall the argument here seems very weak.


**Final Rating After The Rebuttal:**

4: Weak Accept

**Justification Of The Final Rating:**

I believe adding quantitative results and more related work and baseline comparisons strengthens the paper quite a bit and removes the main concerns I had with this paper. Overall, as previously stated I think the paper is nicely structured and written. My only concern is with the novelty of this 'methodological development' paper, which is the translation of a previous method. Consequently, I would rather see it as an application/validation paper. I think the promise of releasing the dataset is great.  In total, I think it would make a good contribution to the conference, and am swaying between a strong accept and a weak accept (If possible I would give a "normal" accept). However, given the limited novelty, I will opt for a weak accept.

**Paper Type:**

methodological development

**Questions To Address In The Rebuttal:**

- Compare yourself to previous works and also show the baseline performance of different models.
- For a methodological paper, please show the benefits of SUPN qualitatively (preferably on a dataset for which localization labels are available).

**Special Issue:**

no

---

### Official Review · Reviewer_bRU1 · 2022-01-24

**Confidence:** 4
**Preliminary Rating:** 4
**Recommendation:** Poster

**Summary:**

The paper proposed a method to perform unsupervised anomaly detection using VAE and image uncertainty. The method performs a two-stage learning strategy to first represent healthy images with a VAE assuming diagonal covariance and later continue training to fit a covariance matrix assuming pixel-wise correlation. As the covariance matrix is high dimensional and inefficient to model its inverse directly, the authors perform an approximation through Cholesky decomposition on the positive semi-definite covariance matrix. This overall method is interesting and addresses an issue in the existing UAD method in modelling the pixel-wise relation. The method is evaluated on images of human red blood cells and aims to detect malaria parasites in the samples. The results show comparison between different ways of modelling the covariance on synthetic and real anomalies.

**Strengths:**

The authors propose to model pixel-wise correlation in the covariance in a principled way, which, potentially, can improve the detection accuracy in general, as most existing VAE-based works treat P(X|z) as a Gaussian with diagonal covariance, and yet the image pixels are often not independent. Further investigation is also presented to show the performance on different types of anomalies and compare the effectiveness of different uncertainty modelling choices.

**Weaknesses:**

I have a few concerns about the method and the results.

Major concerns:
1) As the inverse of the covariance is decomposed into LL^T, and L is a lower triangular matrix, although the regularization tries to enforce off-diagonal values to be 0 or close to 0, the outcome may not be a lower triangular matrix. Is this also enforced in some way, if not, how does this affect the strength of the approximation?
2) In Fig. 2, the authors show samples for Smooth Ellipsoid, could the authors provide some details for how a smoothly varying distortion is calculated?
3) For samples with anomalies, could the authors also highlight the anomalies in those images for better illustration?
4) With the current results, we can visually compare the effectiveness of spherical, diagonal and SUPN, but sometimes the visual results are quite similar, could the authors also provide quantitative comparison, such as area under the ROC curve or precision-recall curve?

Minor edit:
Fig.3 .....where bright parts are be interpreted as outlier features.... -> .....where bright parts can be interpreted as outlier features....


**Deanonymize Review:**

no

**Detailed Comments:**

Please see the weaknesses section.

**Final Rating After The Rebuttal:**

4: Weak Accept

**Justification Of The Final Rating:**

The authors have addressed the concerns, and the additional quantitative results have strengthened the method. I also like the further information provided in the appendix. Overall, I would like to suggest a weak accept.

**Paper Type:**

methodological development

**Questions To Address In The Rebuttal:**

Please address the questions raised in the weaknesses section regarding the constraint of the objective function and providing quantitative measures of different modelling choices and annotated images for better comparison.

**Special Issue:**

no

---

### Official Review · Reviewer_Gxxc · 2022-01-25

**Confidence:** 2
**Preliminary Rating:** 3

**Summary:**

This paper proposes an unsupervised approach to anomaly detection in bright-field or fluorescence cell microscopy. This is achieved by building a generative model (VAE) that describes healthy cell images, where the authors additionally model the structure of the predicted image uncertainty, rather than assuming pixel-wise independence in the likelihood function. The authors employ the recently published Structured Uncertainty Prediction Networks approach to enable tractable learning of the uncertainty structure. The authors demonstrate that their proposed approach is more effective for detecting real and synthetic structured image perturbations compared to diagonal Gaussian likelihoods. Figure 2 show that malaria parasites can be highlighted using their method.

**Strengths:**

1. The way that use data distribution to detect malaria parasites is novel.
2. A validation of the benefits of structured uncertainty when identifying synthetic artefacts applied to bright-field microscopy.

**Weaknesses:**

The authors give reliable visual comparisons, but lack sufficient quantitative comparisons to demonstrate that methods that use data distribution to detect parasites are superior to other comparison methods.

**Deanonymize Review:**

no

**Final Rating After The Rebuttal:**

3: Borderline

**Justification Of The Final Rating:**

I think the paper is well structured and written. Applying VAE-based methods to positioning malaria parasites is a relatively novel use in this field, but the method is less innovative, and it is hoped that there will be a better enrichment of experiments to support it.

**Paper Type:**

methodological development

**Questions To Address In The Rebuttal:**

Will distribution disturbances due to other factors also be considered parasites? The method proposed by the authors is not directly related to the parasite itself. I think this way is more like a trick.

**Special Issue:**

no

---

### Meta-Review · Area_Chair_dZue · 2022-02-20

**Recommendation:** Accept (Poster)
**Confidence:** 5

**Metareview:**

After rebuttal, the manuscript receives 2 weak accept, and 2 borderline. The authors have clarified the issues raised by the reviewers. Two reviewers concern the limited novelty. But they also think the proposed work and released dataset will make a good contribution to the conference. Overall, most reviewers are satisfied with the response given by the authors and are glad to see that the quality of the paper has been improved substantially. It reaches the minimum requirement for publication.

---

### Decision · Program_Chairs · 2022-02-28

Accept